# Intrahepatic cholestasis of pregnancy and gestational diabetes: Protocol for a scoping review of associations, risk factors, and outcomes

**Karima Abubakr[1]\*, Clare Kennedy[1], Shahad Al-Tikriti[1], Amy C. O'Higgins[2], Ciara Coveney[3], Mensud Hatunic[4], Mary F. Higgins[1]**

**1** UCD Perinatal Research Centre, University College Dublin Obstetrics and Gynaecology, National Maternity Hospital, Dublin, Ireland, **2** UCD Obstetrics and Gynaecology, Obstetrics and Gynaecology, Coombe Hospital, Dublin, Ireland, **3** Midwifery, National Maternity Hospital, Dublin, Ireland, **4** Endocrinology, Mater Misercordiae Hospital, Dublin, Ireland

\* Karima.abubakr@ucdconnect.ie

## Abstract

### Background

Intrahepatic cholestasis of pregnancy (IHCP) is a pregnancy-related liver disease associated with adverse pregnancy outcomes, including spontaneous preterm labour, fetal hypoxia, meconium-stained liquor and intrauterine death. In women with IHCP, comorbidities may be associated with a greater risk of stillbirth. Recent studies have suggested that cholestasis of pregnancy may be associated with Gestational Diabetes Mellitus (GDM).

### Objective

This scoping review aims to comprehensively investigate the nature and strength of the association between Intrahepatic Cholestasis of Pregnancy (IHCP) and Gestational Diabetes Mellitus (GDM). The review also seeks to identify common risk factors contributing to the association and explore potential adverse effects associated with the concurrent presence of IHCP and GDM. The findings will inform clinical practice and guide future research initiatives in understanding and managing these pregnancy-related conditions.

### Methods

The scoping review followed the guidelines of Arksey, and O'Malley established in 2005, as well as modifications made to them by Levac et al. in 2010. The PRISMA Scoping review guidance shall be followed in reporting this study. Eight different databases are proposed to search, including Google Scholar, PubMed, Cochrane, Scopus, Embase, CINAHL, the American Diabetes Association, and the Wiley Online Library. Additionally, focused searches within the American Journal of Obstetrics and Gynecology (AJOG) will be conducted, and citation pearl indexing performed.

**Data availability statement:** No datasets were generated or analyzed during the current protocol study. Once the scoping review is completed, relevant data will be published.

**Funding:** The author(s) received no specific funding for this work.

**Competing interests:** The authors have declared that no competing interests exist.

## Introduction

Intrahepatic cholestasis of pregnancy (IHCP) is a pregnancy-related liver disease [1], characterized by pruritus without a primary skin disease, accompanied by elevated maternal bile acid concentrations in the absence of any other medical condition [2]. Symptoms usually appear in the third trimester, but they can appear earlier in pregnancy [3]. The prevalence of IHCP varies by population and is influenced by genetic and environmental factors, with a reported higher prevalence in Chile. In the United Kingdom, the overall incidence was reported at 0.7%, with a higher prevalence among Asian Indians and Pakistanis (1.24% and 1.46%, respectively) than white Caucasians (0.62%) [2,4,5]. It is associated with adverse pregnancy outcomes, including spontaneous preterm labour, fetal hypoxia, meconium-stained liquor, and stillbirth, where severe IHCP patients are at higher risk of adverse pregnancy outcomes [6–14], supporting increased surveillance and support for pregnancies affected by severe IHCP. In the 2022 guideline, the Royal College of Obstetricians and Gynecologists considered a new IHCP-diagnosed criterion based on peak random total bile acid concentration of ≥19 micromol/L in pregnant women who have pruritus on skin that appears normal [2].

The timing of birth, based on the IHCP severity, is currently the primary treatment approach, as there have been no treatments that have been shown to enhance pregnancy outcomes, and medications that improve maternal pruritus are of insufficient benefit [2]. Despite studies [9,15,16] supporting the use of ursodeoxycholic acid as a first-line treatment, evidence from randomized controlled trials shows that there is no reduction in adverse perinatal outcomes in women given ursodeoxycholic acid compared to women in the placebo group [17,18]. Active care in IHCP usually refers to planned birth, with timing dependent on multiple factors.

Recent studies have suggested that cholestasis during pregnancy may be associated with gestational diabetes (GDM) [19–23]. Clinical studies also suggest a link between IHCP and GDM, though this may be further complicated by different methods of diagnosing GDM. Studies from Sweden [24], China [25], Australia [26], the United States [27], Poland [28], a more recent study from China [29], a Danish study [30], and meta-analysis studies that also found that the IHCP is closely related to increased risk of GDM during pregnancy [31–33].

The co-occurrence of IHCP and GDM poses dual challenges, increasing the likelihood of adverse outcomes for both women and fetuses, including the consequence of intrauterine death [13]. Additionally, while contemplating the combination of IHCP and GDM, it is critical to examine the possible impact on additional complications, such as the requirement for births by cesarean due to macrosomia as well as adverse perinatal outcomes; such issues might impact recommendations about the precise timing of a planned birth [2].Understanding the complicated link between IHCP and GDM is critical to developing effective management plans and ensuring the well-being of both mothers and infants.

In a recent systematic review and meta-analysis conducted in 2021 [33], the prevalence of the association between Intrahepatic Cholestasis of Pregnancy (IHCP) and Gestational Diabetes Mellitus (GDM) and the risk of intrauterine death was comprehensively investigated. This meta-analysis systematically analyzed published articles from January 1, 2010, to November 27, 2020. However, since the conclusion of this meta-analysis, new research has emerged, delving deeper into the intricate relationship between IHCP and GDM. This scoping review aims to provide an updated overview, with a specific emphasis on exploring risk factors associated with this dual pathology and elucidating the maternal and fetal outcomes, which were not a focus of the previous systematic review.

There is currently no recommendation to screen for GDM in cases of IHCP, and more research is required to determine the relationship between GDM and IHCP. The proposed scoping review aims to provide a valuable resource to health practitioners by providing

insights into the current state of knowledge on the interaction between IHCP and GDM. Furthermore, it intends to bridge the existing research gap by highlighting areas that require further exploration, ultimately contributing to the progress of our understanding of this delicate interaction for the benefit of mother and fetal health.

## Methods

### Overview

The scoping review will follow the guidelines of Arksey and O'Malley [34] established in 2005, as well as modifications made to them by Levac et al. [35] in 2010, with the following steps: (1) identifying research questions; (2) identifying relevant articles; (3) selecting articles; (4) charting data; (5) presenting the results, discussions, and conclusion; and (6) expert consultation. Study materials and data are shared on the OSF in the Centre for Open Science: https://osf.io/fkpb5/?view_only=5a05306389e24b1097c0bc3f929bfed1.

The PRISMA-P (Preferred Reporting Items for Systematic review and Meta-Analysis Protocols) 2015 guidelines shall be followed in reporting this study (see S1 File).

### Identifying research question

To identify the research question, the search strategy and specific terms used were the Population-Concepts-Context (PCC) strategy proposed by Levac et al. in Table 1. Based on this PCC strategy, the scoping review question is: What is the extent of the published literature concerning the association between Intrahepatic Cholestasis of Pregnancy (IHCP) and Gestational Diabetes Mellitus (GDM), and what insights does it offer into the risk factors and outcomes associated with this relationship?

### Identifying relevant articles

Eight different databases will be chosen to search, including Google Scholar, PubMed, Cochrane, Scopus, Embase, CINAHL, the American Diabetes Association, and the Wiley Online Library. Additionally, focused searches within the American Journal of Obstetrics and Gynecology (AJOG) will be conducted, and the reference lists of all included papers will be reviewed for the potential inclusion of further research matching eligibility requirements. This strategy aims to ensure a thorough analysis of diverse sources, including various study designs and contexts, to completely synthesize information on the topic.

All original peer-reviewed research methods, including qualitative, quantitative, and mixed-methods studies, reporting on the association between IHCP and GDM in pregnant populations, published from 2010 to the present will be included. The search will start from 2010 due to the change in the diagnosis of GDM following the adoption of the International Association of Diabetes and Pregnancy Study Groups (IADPSG) guidelines [36], these guidelines introduced revised criteria for diagnosing gestational diabetes, increasing diagnosis rates, providing a more accurate representation of the prevalence and characteristics of the condition, and standardizing the reference of research articles [37]. Animal-model studies, case reports, and reviews, as well as non-English articles will be excluded.

**Table 1. Framework of research question.**

| Population (P) | Concept (C) | Context (C) |
|---|---|---|
| Pregnant people with intrahepatic cholestasis of pregnancy (IHCP) and gestational diabetes (GDM). | The extent and nature of the association between IHCP and GDM, with a particular focus on clarifying the risk factors and outcomes associated with this association. | Antenatal care. |

The proposed keywords, free text and MESH terms for the search were chosen after a pilot search and review of previously identified papers to develop a search strategy as follows: (("intrahepatic cholestasis of pregnancy" OR "Pregnancy related cholestasis " OR "Obstetric Cholestasis" OR '' jaundice of pregnancy" OR '' prurigo gravidarum" OR '' obstetric hepatosis" OR '' hepatosis gestationalis") AND ("Pregnancy Induced Diabetes" OR "Gestational Diabetes" OR "Glucose Tolerance Test" OR "Glucose Challenge Test"))

## Selecting the articles

Articles will be reviewed based on title, then abstract, then full text. Authors will be contacted for clarification as required. After exporting search results, duplicated articles will be deleted. Rayyan, an online tool allowing this review [38], will be used with the lead author performing the review and two other authors performing separate reviews. If there is no consensus between authors on inclusion or exclusion of specific studies, this will be resolved by discussion or reference to another author. Flow of studies through the searching and review stages are shown in Fig 1.

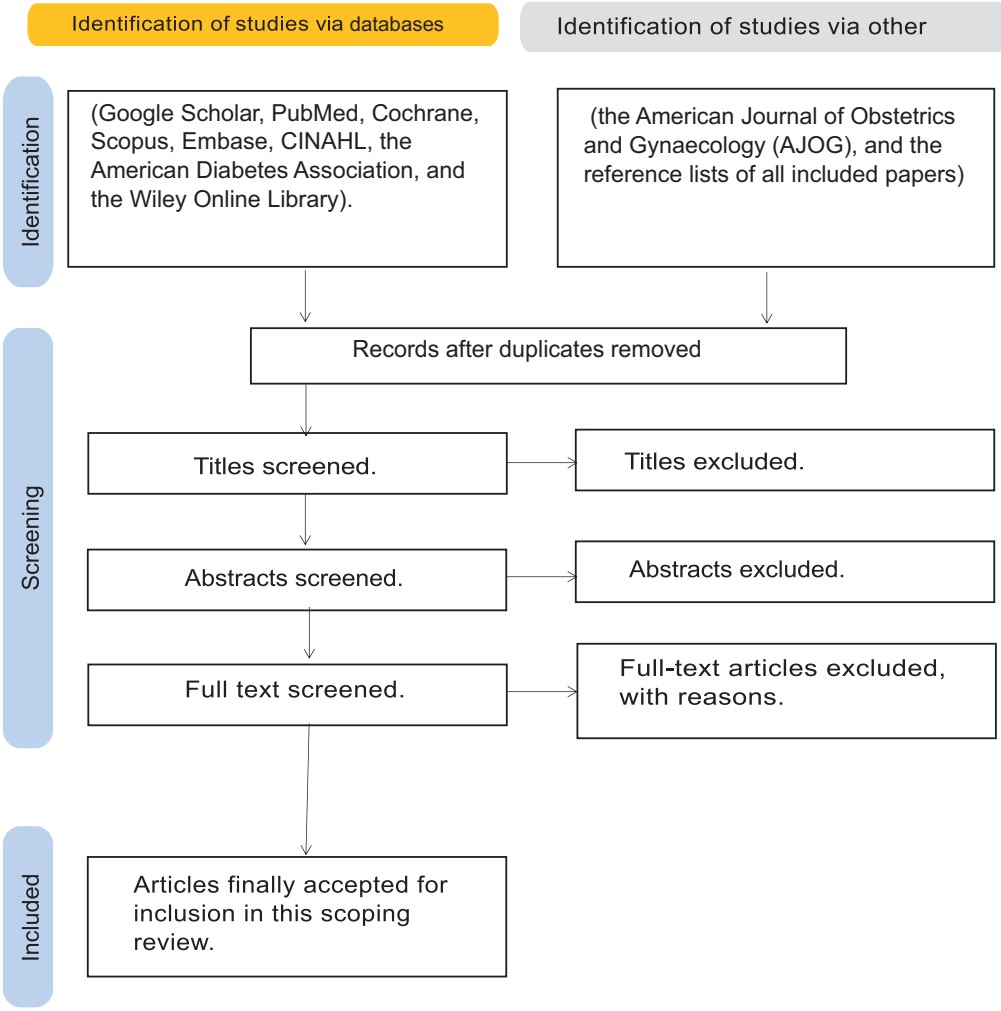

**Fig 1. PRISMA-ScR flow diagram showing the steps of literature search, screening, and study selection procedures for the review.**

## Charting the data

An initial pilot data extraction form was piloted within the research team and then modified after piloting with ten papers that could potentially be included in the scoping review. Data was selected for initial extraction, noting that the data extraction form is susceptible to change and modification in accordance with the purpose of this scoping review question.

The proposed data extraction form included research parameters (*first author, year of publication, study design, country location of the study, continent, duration of collection of study data, duration of collection of individual patient data, sample size, objective*). In investigating the association, the GDM parameters (screening criteria, screening risk factor criteria, screening method, diagnostic criteria), diagnostic criteria for IHCP, prevalence or incidence rates of IHCP and GDM, measures of association (e.g., odds ratio, relative risk) between IHCP and GDM, and the association's confidence intervals, or p-values, will be extracted.

The identification and comparative study of risk factors related to the coexistence of IHCP and GDM will be included, specifically maternal age, Body mass index (BMI), family history of diabetes, ethnicity, and gestational age at diagnosis.

Further data, maternal and fetal outcomes associated with this coexistence will be documented, such as higher risks of preterm labor and mode of delivery, as well as neonatal problems.

## Collate, summarize, and report the results

The aim of a scoping review is to create, summarize, and report the results in three steps. Initially, descriptive numerical analysis will be displayed to encompass some of the included articles' features. Furthermore, the literature's strengths and limitations will be discovered through theme analysis of the studies included in the scope of the research. The last stage is an assessment of the findings' implications for future observations, practices, and policies.

## Expert consultation

While it was initially proposed [34] that consultation could be optional in doing scoping research, a more recent proposal [35] contends that it adds methodological rigor and should be regarded as a compulsory component. The expert consultation stage is an important quality assurance technique that ensures the scoping review properly collects the relevant literature and insights within the intended scope.

Expert consultation will be sought throughout the scoping review process to ensure the outcomes' comprehensiveness and accuracy. The research study team included multidisciplinary team members who provide care to pregnant women who may develop IHCP, GDM, or both during pregnancy, and each will discuss the review with the team members. We aim to run a parallel research study that may investigate women's experience of developing IHCP and, as part of this, women will be asked to identify areas they consider important in the research. Finally, advocacy groups on social media representing women with IHCP and GDM have been approached to give feedback on this scoping review question.

## Ethical considerations

This scoping evaluation used publicly accessible data and did not require ethical approval.

## Discussion

This scoping review seeks to offer a complete assessment of the current knowledge about the relationship between IHCP and GDM by addressing primary and secondary objectives. This

review, however, is limited to IHCP with GMD and excludes other diabetes types. Additionally, it is important to note that this review is limited to articles published in the English language.

## Dissemination

The findings of the scoping review will be discussed through regional and national scientific conference proceedings and presentations, stakeholder meetings, and publishing in a peer-reviewed journal.

## Supporting information

**S1 File. PRISMA-P checklist.** Title: Preferred Reporting Items for Systematic review and Meta-Analysis Protocols (PRISMA-P) 2015 statement. Legend: This file contains the PRISMA-P (Preferred Reporting Items for Systematic review and Meta-Analysis Protocols) 2015 checklist: recommended items to address in a systematic review protocol. The checklist is available at https://doi.org/10.1186/2046-4053-4-1 [39].
(PDF)

## Acknowledgments

This scoping review is contributing towards a thesis paper for an MD by research at University College Dublin (UCD) by KA.

## Author contributions

**Conceptualization:** Karima Abubakr.

**Data curation:** Karima Abubakr, Clare Kennedy, Shahad Al-Tikriti.

**Formal analysis:** Karima Abubakr.

**Investigation:** Karima Abubakr.

**Methodology:** Karima Abubakr, Clare Kennedy, Shahad Al-Tikriti, Amy C. O'Higgins, Ciara Coveney, Mensud Hatunic.

**Project administration:** Karima Abubakr.

**Resources:** Karima Abubakr.

**Software:** Karima Abubakr.

**Supervision:** Mary F. Higgins.

**Validation:** Karima Abubakr, Clare Kennedy, Shahad Al-Tikriti, Amy C. O'Higgins, Ciara Coveney, Mensud Hatunic, Mary F. Higgins.

**Visualization:** Karima Abubakr.

**Writing – original draft:** Karima Abubakr.

**Writing – review & editing:** Karima Abubakr, Clare Kennedy, Shahad Al-Tikriti, Amy C. O'Higgins, Ciara Coveney, Mensud Hatunic, Mary F. Higgins.

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
