## [Decision Letter · Decision Letter 0]

24 Jan 2025

Intrahepatic Cholestasis of Pregnancy and Gestational Diabetes: Protocol for a Scoping Review of Associations, Risk Factors, and Outcomes

PONE-D-24-06123

Dear Dr. Abubakr,

We’re pleased to inform you that your manuscript has been judged scientifically suitable for publication and will be formally accepted for publication once it meets all outstanding technical requirements.

Kind regards,

Consolato M. Sergi

Academic Editor

PLOS ONE

- "https://www.fasgo.org.ar/images/colestasis_intrahepatica_del_embarazo.pdf""

In your revision ensure you cite all your sources (including your own works), and quote or rephrase any duplicated text outside the methods section. Further consideration is dependent on these concerns being addressed.

Reviewers' comments:

Reviewer's Responses to Questions

**Comments to the Author**

1. Does the manuscript provide a valid rationale for the proposed study, with clearly identified and justified research questions?

Reviewer #1: Yes

Reviewer #2: Yes

2. Is the protocol technically sound and planned in a manner that will lead to a meaningful outcome and allow testing the stated hypotheses?

Reviewer #1: Yes

Reviewer #2: Yes

3. Is the methodology feasible and described in sufficient detail to allow the work to be replicable?

Reviewer #1: Yes

Reviewer #2: No

4. Have the authors described where all data underlying the findings will be made available when the study is complete?

Reviewer #1: Yes

Reviewer #2: Yes

5. Is the manuscript presented in an intelligible fashion and written in standard English?

Reviewer #1: Yes

Reviewer #2: No

6. Review Comments to the Author

You may also provide optional suggestions and comments to authors that they might find helpful in planning their study.

Reviewer #1: The protocol demonstrates strong technical solidity through its comprehensive and methodologically sound approach, following established guidelines like Arksey and O’Malley and PRISMA-ScR. The planned statistical analysis is robust, aiming to explore associations and risk factors with precise measures such as odds ratios and confidence intervals, ensuring accurate interpretation of data. The presentation is clear and well-structured, with detailed steps for data collection and analysis, enhancing the reliability and transparency of the review process.

Reviewer #2: The authors conducted a scoping review to investigate the nature and strength of the association between Intrahepatic Cholestasis of Pregnancy (IHCP) and Gestational Diabetes Mellitus (GDM), identify common risk factors to the association, and explore potential negative effects in line with the concurrent presence of IHCP and GDM. Despite a clearly delineated methodology, the scoping review lacks an overview of the included sources of evidence, a thorough discussion of the findings, and an acknowledgment of the eventual limitations.

The authors included various sources of evidence. They searched multiple databases and consulted the grey literature. Furthermore, they screened titles and abstracts, and a proper critical appraisal of individual sources of evidence was conducted. The use of independent screening by different researchers added rigor to the methodology.

However, no actual overview of the data was presented in this scoping review. The authors need to describe the data from the set of sources finally included in the scoping review. For instance, the overall picture of the extracted research parameters—such as their distribution, range, and averages—is not provided. This includes presenting data on measures of association and key variables like maternal age, body mass index, and other risk factors related to the coexistence of IHCP and GDM. Understanding these distributions and trends is crucial to provide a comprehensive view of the evidence base.

Furthermore, the results should be presented comprehensively, along with a discussion of their implications and a clear acknowledgment of limitations. The authors should also indicate the findings' significance for future research and clinical practice.

Figure 1: Selection of Sources of Evidence

The number of sources screened and assessed for eligibility was not mentioned. Indicate in the text boxes the number of sources identified, screened, and finally included in the scoping review. The number of records remaining after duplicate removal should also be provided.

Funding Sources

The authors need to describe the sources of funding for both (1) the included sources of evidence and (2) the scoping review itself. As this review is intended to contribute to a thesis paper, it is essential to specify how the included articles and other sources of evidence were funded. This transparency will enhance the review's credibility and provide context for potential biases in the included studies.

7. PLOS authors have the option to publish the peer review history of their article (what does this mean? ). If published, this will include your full peer review and any attached files.

**Do you want your identity to be public for this peer review?** For information about this choice, including consent withdrawal, please see our Privacy Policy .

Reviewer #1: **Yes: ** Bhargav Koyani

Reviewer #2: **Yes: ** Florent Joseph Feulefack

---

## [Editor Report · Acceptance letter]

PONE-D-24-06123

PLOS ONE

Dear Dr. Abubakr,

I'm pleased to inform you that your manuscript has been deemed suitable for publication in PLOS ONE. Congratulations! Your manuscript is now being handed over to our production team.

Kind regards,

on behalf of

Professor Consolato M. Sergi

Academic Editor

PLOS ONE